# Temporal Dynamics of *Anaplasma marginale* Infections and the Composition of *Anaplasma* spp. in Calves in the Mnisi Communal Area, Mpumalanga, South Africa

**DOI:** 10.3390/microorganisms11020465

**Published:** 2023-02-13

**Authors:** S. Marcus Makgabo, Kelly A. Brayton, Louise Biggs, Marinda C. Oosthuizen, Nicola E. Collins

**Affiliations:** 1Department of Veterinary Tropical Diseases, Faculty of Veterinary Science, University of Pretoria, Private Bag X04, Onderstepoort 0110, South Africa; 2Department of Veterinary Microbiology and Pathology, Washington State University, Pullman, WA 99164, USA; 3Department of Production Animal Studies, University of Pretoria, Onderstepoort 0110, South Africa

**Keywords:** *Anaplasma marginale*, *msp1α* gene, wildlife–livestock interface, genotyping, tick-borne diseases, detection, diagnosis

## Abstract

Bovine anaplasmosis, caused by *Anaplasma marginale*, is one of the most important tick-borne diseases of cattle. *Anaplasma marginale* is known to be present in the Mnisi community, Mpumalanga Province, with frequent cases of anaplasmosis reported. This study investigated the infection dynamics in calves (*n* = 10) in two habitats in the study area over 12 months. A duplex real-time PCR assay targeting the *msp1β* gene of *A. marginale* and the *groEL* gene of *A. centrale* confirmed the presence of *A. marginale* in five calves in a peri-urban area from the first month, but in only two calves at the wildlife–livestock interface and only after six months. These results were confirmed by 16S rRNA microbiome analysis. Over 50 *A. marginale msp1α* genotypes were detected in the calves along with five novel Msp1a repeats. Calves in the peri-urban area were more likely to be infected with *A. marginale* than calves in the wildlife–livestock interface. Cattle management, acaricide treatment, and cattle density could explain differences in infection prevalence in the two areas. Our results revealed that most calves were superinfected by distinct *A. marginale* strains within the study period, indicating continuous challenge with multiple strains that should lead to robust immunity in the calves and endemic stability in the area.

## 1. Introduction

Bovine anaplasmosis is a tick-borne disease (TBD) caused by the obligate intracellular pathogen, *Anaplasma marginale* [1]. Bovine anaplasmosis occurs predominantly in cattle; however, infection can also occur in ruminants such as sheep, goats, African antelopes, Cape buffalo, and some species of deer [1].

Bovine anaplasmosis caused by *A. marginale* is prevalent throughout the world occurring in Africa, the Americas, Asia, Australia, the Caribbean, and Europe [2]. The disease is amongst the three most economically important TBDs of cattle resulting in mortality and morbidity, decreased milk and meat production, and expensive control measures [3,4,5,6]. The economic impact of bovine anaplasmosis in South Africa has been estimated at approximately R115 million ($US9.6 million) per year due to mortalities [7], but this does not take into account costs associated with treatment and control. In other parts of the world, costs arising from anaplasmosis have been estimated from USD 300 to USD 800 million [2]. Clinical signs caused by infection with *A. marginale* are characterized by fever, progressive anaemia, weight loss and abortion, as well as icterus that may result in mortality. The closely related organism, *A. centrale*, usually causes asymptomatic infections and is currently used as a vaccine for cattle in South Africa [8,9,10]. Animals under one year of age are usually asymptomatic to infection with *A. marginale* [11,12]. However, older animals are more likely to react severely and fatally upon challenge [1].

Biological transmission of *A. marginale* to naïve cattle occurs by feeding ticks, while mechanical transmission occurs by biting flies or blood-contaminated instruments [13,14]. Transplacental transmission of *A. marginale* has also been reported [1,15]. Although the transmission of *A. marginale* in South Africa has not been extensively studied, five tick species, *Rhipicephalus decoloratus*, *R. microplus*, *R. evertsi evertsi*, *R. simus*, and *Hyalomma marginatum rufipes*, have been shown experimentally to transmit anaplasmosis and could therefore account for the widespread distribution of the disease [4,10]. *Rhipicephalus decoloratus* has long been regarded as the main vector [10], with *R. microplus* increasing in importance as a vector due to its recent spread into most South African provinces [16,17].

The genetic diversity of *A. marginale* from many countries around the world has been characterized using a genotyping method based on sequence analysis of the single-copy *msp1α* gene that codes for the major surface protein 1a (Msp1a) [18,19,20]. Msp1a is regarded as a determining marker for *A. marginale* transmission between cattle and ticks as it has been shown to be involved in the adhesion of the pathogen to tick cells and bovine erythrocytes [1]. The genotyping method uses differences in the number and sequence of tandem repeats located at the N-terminus of the Msp1a protein to differentiate between strains. The *msp1α* genotyping method was first described in 1990, and since then >300 genotypes have been reported worldwide [21,22]. In South Africa, a diversity of *A. marginale* genotypes has also been identified [19,20,23,24,25]. 

The presence of single *msp1α* genotypes in infected cattle is a well-documented phenomenon [13,26], but infection with multiple *A. marginale* strains (superinfection) has been less well studied. More recently, both co-infection and superinfection of cattle with multiple genetically distinct strains of *A. marginale* have been shown to be important drivers of *A. marginale* infection [27,28,29,30]. Co-infection and superinfection were recently shown to drive the development of complex infection with *A. marginale* under natural transmission conditions in Ghana [31].

In the South African context, bovine anaplasmosis is currently widespread and endemic throughout the cattle-farming areas in all South African provinces, except for the Northern Cape, where the vector is mostly absent [4,6,24,25]. Data collected through the Health and Demographic Surveillance System in Livestock (HDSS) established in the study area of the Mnisi community, indicate the presence of *A. marginale* in cattle, with frequent bovine anaplasmosis cases reported at villages that abut provincial and private game reserves (the wildlife–livestock interface) [32]. The Mnisi community is a sprawling area that provides an opportunity to study natural *A. marginale* infection dynamics at both more densely populated peri-urban villages and at villages at the wildlife–livestock interface [33].

To understand *A. marginale* strain diversity, infection dynamics, and the frequent nature of clinical cases of anaplasmosis in the Mnisi community, ten calves were examined from birth for a period of 12-months in a peri-urban area and at a wildlife–livestock interface.

## 2. Materials and Methods

### 2.1. Ethical Consideration

The study was carried out in strict accordance with the conditions and guidelines of the Animal Ethics Committee of the Faculty of Veterinary Science (reference number: V041-16)**.** Permission to perform the study under Section 20 of the Animal Disease Act of 1984 was granted by the South African Department of Agriculture, Forestry and Fisheries (currently Department of Agriculture, Land Reform and Rural Development) (reference number: 12/11/1/1/6).

### 2.2. Study Area

The Mnisi community (24.8205° S, 31.1710° E) is situated in the north-eastern corner of the Bushbuckridge Municipality, Mpumalanga Province, South Africa (Figure 1). The community shares 75% of its boundary with adjacent wildlife areas, including the Andover and Manyeleti provincial game reserves and the Timbavati and Sabi Sand private game reserves. There are no fences between these reserves, including the Kruger National Park (KNP), such that game can freely roam between them. Livestock farming is the main agricultural activity in the area with more cattle than any other livestock species. The project was conducted in three villages, Eglington, Utha A, and Dixie. Eglington village is in a peri-urban area, while Utha A and Dixie are located at the wildlife–livestock interface close to the border with Manyeleti provincial game reserve.

Over 40,000 people live in the Mnisi community. Eglington village is a peri-urban area, situated approximately 11.5 km away from the Manyeleti Game Reserve, 12.1 km from the Andover Game Reserve, and 15.1 km from the Timbavati Game Reserve. Each day, cattle herders collect the cattle from the owners’ homes where they are kept in kraals overnight, and they are taken to a fully protected and fenced cattle grazing camp, where the chosen calves grazed during the study period. The Eglington cattle grazing camp is located approximately 16 km away from Manyeleti Game Reserve, 13 km from Andover Game Reserve, and 15.1 km from Timbavati Game Reserve. Utha A and Dixie villages are only 2 km apart and are located close to the wildlife–livestock interface being, respectively, approximately 2.4 km and 0.5 km away from the Manyeleti Game Reserve. In this area, cattle grazing camps are located adjacent to the Manyeleti Game Reserve and cattle are often seen grazing alongside wildlife separated only by the game fence, which is the only barrier between livestock and the abundant wildlife populations in the game reserves. Due to the study area being in the foot and mouth disease (FMD) protection zone, animals can move between villages in the zone only with permission from the state vet, however, trading of livestock out of the zone is not permitted. There is therefore little animal cross-traffic between villages. The community is characterized by a human health centre, animal health clinic, and shopping complex in Hluvukani, where people from the different villages gather. There is human cross-traffic in the study area, with villagers, commuters, researchers, and veterinarians travelling freely between villages.

Due to the study area being in the FMD protection zone, and the proximity of wildlife species, which harbour and facilitate the spread of ticks and tick-borne diseases between wildlife, livestock and humans, comprehensive disease surveillance measures are implemented in the area by local veterinary services, mainly in the form of cattle dip tanks built throughout the region, which every cattle herd must visit for dipping and FMD inspection once a week. The dip consists of the Delete^®^ X5 acaricide which is used on cattle, sheep, and goats, for the prevention and treatment of ectoparasite infestation. The farmers in the Mnisi community do not vaccinate their cattle against bovine anaplasmosis.

### 2.3. Animals 

Ten local mixed breed *Bos taurus* calves (0–1 months of age, 6 males and 4 females) were monitored for a period of one year. Three of the ten calves were situated in Utha A (with a total of 715 cattle and a cattle density of 128 cattle/km^2^) and two were in Dixie (with a total of 137 cattle and cattle density of 27 cattle/km^2^); these two villages are located approximately 2.4 km and 0.5 km away, respectively to the wildlife–livestock interface. The remaining five cattle were based in Eglington village (with a total of 1009 cattle and a cattle density of 194 cattle/km^2^); this is a peri-urban area, located 11.5 km away from the border with Manyeleti Game Reserve. The local veterinary services used the following different methods of acaricide treatment in the two areas: the plunge method of dipping cattle was used at Eglington (the peri-urban area), as well as Utha A (wildlife–livestock interface), while the hand spraying method was used at Dixie village (wildlife–livestock interface) due to water-shortages. The study required farmers with a relatively small herd of cattle who do not dip their cattle privately.

### 2.4. Study Design and Sample Collection

This longitudinal study was conducted between November 2016 (when the calves were 0–1-month-old) and October 2017. Whole blood samples were collected in 10 mL Vacutainer^®^ ethylenediaminetetraacetic acid (EDTA) tubes from the ten calves once a month for 12 months according to the 12 time-point collection timeline (Figure 2).

### 2.5. Genomic DNA Extraction and Quantitative Real-Time PCR (qPCR) Assay

Genomic DNA was extracted from the samples collected from all time-points using the QIAamp DNA Blood Mini Kit (Qiagen, Hilden, Germany) according to the manufacturer’s instructions. DNA was eluted in 100 μL elution buffer and stored at −20 °C. Genomic DNA samples were screened for the presence of *A. marginale* and *A. centrale* using a duplex qPCR assay targeting the *msp1β* gene of *A. marginale* and the *groEL* gene of *A. centrale* [34]. Primers, AM-For (5′-TTG GCA AGG CAG CAG CTT-3′), AM- Rev (5′-TTC CGC GAG CAT GTG CAT-3′) and a probe, AM-Pb (6-FAM-TCG GTC TAA CAT CTC CAG GCT TTC AT-BHQ1) were used to amplify and detect a 95 bp fragment of the *msp1β* gene of *A. marginale* while primers, AC-For (5′-CTA TAC ACG CTT GCA TCT C-3′), AC-Rev (5′-CGC TTT ATG ATG TTG ATG C-3′) and probe AC-Pb (LC610-ATC ATC ATT CTT CCC CTT CCC CTT TAC CTC GT-BHQ2) were used to amplify a 77 bp fragment of the *groEL* gene of *A. centrale*. Reactions were performed in a 20 µL final reaction volume comprising 4 µL FreshStart Taqman mix (Roche Diagnostics, Midrand, South Africa), 0.5 µL UDG, 0.6 µM of the *A. marginale*-specific primers, 0.9 µM of the *A. centrale*-specific primers, 0.2 µM of each probe, and 2.5 µL of template DNA (approximately 200 ng). The duplex assay was performed on a LightCycler v2 (Roche Diagnostics, Mannheim, Germany), using the thermal cycling conditions described previously [34]. Positive control for the *A. centrale* assay was DNA extracted from the *A. centrale* vaccine strain obtained from Onderstepoort Biological Products (OBP), Pretoria, South Africa. 

Field sample C14 (obtained from cattle in the Mnisi Community area, Mpumalanga) was used as the positive control for the *A. marginale* assay. Nuclease-free water was used as a negative control for the assay. Results were analyzed using the LightCycler Software version 4.0 (Roche Diagnostics, Mannheim, Germany). Samples were run in triplicate per time-point to ensure reproducibility and repeatability of the results. A published linear range of detection and assay efficiency [25,34] were used to quantify the level of *A. marginale* rickettsaemia which was expressed as infected red blood cells (iRBC) per mL of blood.

### 2.6. Amplification, Cloning and Sequencing of the A. marginale Msp1α Gene

The repeat-containing variable region of the *A. marginale msp1α* gene was amplified using primers 1733F (5′-TGT GCT TAT GGC AGA CAT TTC C-3′) and 2957R (5′-AAA CCT TGT AGC CCC AAC TTA TCC-3′) [35]. Amplifications were performed in a 25 µL final reaction volume and consisted of 1x Phusion Flash High-Fidelity PCR Master Mix (Thermo Fisher Scientific, Johannesburg, South Africa), 0.5 µM of each primer, and 2.5 µL of template DNA (approximately 200 ng). The thermal cycling parameters used were modified from those previously reported [35] and comprised a pre-PCR denaturation at 94 °C for 3 min and *Taq* activation at 98 °C for 10 s, followed by 30 cycles of 98 °C for 1 s, 69.1 °C for 5 s, and 72 °C for 18 s, and a final extension at 72 °C for 1 min. PCR products were analysed by electrophoresis on a 1.5% agarose gel (1 × TAE buffer, pH 8.0), stained with ethidium bromide and viewed under UV light. All positive PCR products were purified using the Omega Bio-tek^®^ DNA purification kit (Whitehead Scientific, Modderfontein, South Africa) according to the manufacturer’s instructions. Purified PCR products were cloned into pJET 1.2 (Thermo Fisher Scientific, Johannesburg, South Africa). Recombinant clones were screened by colony PCR using vector specific primers, pJET1.2F and pJET1.2R; clones which yielded a product of 610 bp or greater were selected for sequencing. Fifteen recombinant clones per calf per time-point were sequenced bidirectionally on an ABI Prism 3100 Genetic Analyzer (Applied Biosystems, Foster City, CA, USA) at Inqaba Biotechnical Industries, Pretoria or at the Central Analytical Facility, Stellenbosch University. *Anaplasma marginale msp1α* nucleotide sequences generated in this study were named according to a naming to a proposed naming scheme [22] and deposited in GenBank under accession numbers OQ384772–OQ384912 and are also available under BioProject accession number PRJNA929355.

### 2.7. Characterization of A. marginale Msp1a Repeats and Msp1α Genotypes

Msp1a sequences were trimmed, assembled, edited, and aligned using CLC Genomics Workbench 20.0.4 (Qiagen, https://digitalinsights.qiagen.com/, accessed on 17 January 2023). The RepeatAnalyzer command line software tool [22] was used to identify, store, curate, and analyse Msp1a repeats and *A. marginale msp1α* genotypes. Novel repeats that were not recognized by RepeatAnalyzer were designated UP37 to UP42. The Msp1a repeat structure determines the *msp1α* genotype of a strain.

### 2.8. 16S rRNA Gene Amplification and PacBio Sequencing

In order to determine the composition and diversity of *Anaplasma* species present in the ten calves by T12, the full-length 16S rRNA gene (V1–V9 variable regions) was amplified in triplicate from the ten DNA samples collected at T12 using modified barcoded 16S rRNA gene specific primers, 27F: (5′-AGR GTT YGA TYM TGG CTC AG-3′) and 1492R: (5′-RGY TAC CTT GTT ACG ACT T-3′) recommended for the PacBio Sequel II sequencing instrument (Pacific Biosciences, Menlo Park, CA, USA) [36,37]. Reactions were performed in triplicate in a final volume of 25 µL containing 1 X Phusion Flash^®^ High Fidelity Master Mix (Thermo Fisher Scientific, South Africa), 0.15 µM of each forward and reverse primer, and 5 µL of DNA (approximately 400 ng). DNA extracted from the *A. centrale* vaccine strain (Onderstepoort Biological Products, South Africa) was used as a positive control and molecular grade water as a negative control. Cycling conditions included 98 °C for 30 s, followed by 35 cycles of 98 °C for 10 s, 60 °C for 30 s, and 72 °C for 30 s and a final extension at 72 °C for 10 min. Amplicons were visualized under UV light after electrophoresis on a 1.5% agarose gel stained with ethidium bromide. Amplicons were purified using the QIAquick^®^ PCR purification kit (Qiagen) according to the manufacturer’s instructions and submitted to the Genomics Sequencing Core at Washington State University, Pullman, WA, USA for circular consensus sequencing (CCS) on the PacBio (Pacific Biosciences, Menlo Park, CA, USA) platform.

### 2.9. Analysis of Anaplasma 16S rRNA Sequences Identified by Microbiome Sequencing

The 16S rRNA amplicon sequence data was curated and filtered using SMRT Link software 8.0 according to a minimum barcode score of 70 and 99% precision. Final Fasta and Fastaq data sets were analyzed using the Ribosomal Database Project (RDP) 16S classifier [38,39] for *Anaplasma* genus level classification of the sequences with a 95% confidence interval. Sequences were further classified to the *Anaplasma* species level using a customized NCBI BLASTn database of all known and published *Anaplasma* spp. sequences downloaded from GenBank using the command line application. Sequences were further filtered and excluded based on sequence length (minimum of 1275 bp), quality, and sequence identity in Microsoft Excel [39,40]. Since some distinct *Anaplasma* species are known to have more than 98.7% shared sequence identity, and *A. platys*, *Anaplasma* sp. Mymensingh, “*Candidatus* Anaplasma camelii” and *Anaplasma* sp. Omatjenne share more than 99.5% 16S rRNA gene sequence identity, it is not possible to resolve these organisms to species level [40]. Thus, only 16S rRNA sequences that were identical to previously published sequences were classified to species level; the remainder were reported as putative novel *Anaplasma* species and/or genotypes. The raw microbiome data from the ten calves is available at the Sequence Read Archive (SRA) with BioProject accession number PRJNA929355.

### 2.10. Sequence and Phylogenetic Analysis

The *Anaplasma* 16S rRNA gene sequences identified by microbiome sequencing were aligned with reference sequences from GenBank. *Anaplasma* 16S rRNA sequences from representative genome sequences as well as the most closely related sequences from cattle and other ruminants in South Africa and worldwide, as identified by BLAST analysis, were selected to construct the phylogenetic tree. *Anaplasma* 16S rRNA sequences from wildlife were included [41]. The extent of sequence variation was analysed using CLC Genomics Workbench (Qiagen). Alignments were further trimmed using Bioedit 7 [42]. Jmodel test 1.3 [43] predicted the HKY85 (Hasegawa–Kishino–Yano, 85) evolutionary model [44,45,46] as the best fit model for the 16S rRNA gene sequences. Phylogenetic trees for the 16S rRNA gene were constructed using the neighbor-joining and maximum likelihood (ML) method in MEGA 7 with bootstrap analysis using 1000 replicates to estimate the confidence levels of the tree branches [47], as well as Bayesian inference in Mr Bayes 3.2 [48]. The *Anaplasma* 16S rRNA nucleotide identified in this study were deposited in GenBank under accession numbers OQ348128-OQ348132, with BioProject accession number PRJNA929355.

## 3. Results

The *A. marginale* and *A. centrale* duplex qPCR was used to determine the presence of *Anaplasma* species in the calves. *Anaplasma marginale* was detected in seven of the 10 (70%) calves recruited to the study. Of the seven calves that tested positive for *A. marginale*, five were in the peri-urban area (Eglington village), while only two were located at the wildlife–livestock interface, both in Utha A village. 

Four of the calves in the peri-urban area were already infected with *A. marginale* at the first time point (T1), and by the second time-point (T2), all five calves in this area tested positive (Figure 3A). The two calves that tested positive for *A. marginale* at the wildlife–livestock interface became infected only at T7 and T8, while the remaining three calves at the wildlife–livestock interface were either infected at levels below the detection limit of the assay (250 copies per reaction) or were not infected with *A. marginale* at (Figure 3B). *Anaplasma centrale* was not detected in any of the calves.

The levels of *A. marginale* infection fluctuated over the course of the 12-month study period, exhibiting the cyclic rickettsaemia known to occur in persistently infected animals [2,49]. In calves in the peri-urban area, the rickettsaemia ranged from 4 × 10^6^ to 3 × 10^9^ iRBC/mL from time of infection to a year. The levels of rickettsaemia in the two calves at the wildlife–livestock interface ranged from 2 × 10^6^ to 2 × 10^7^ iRBC/mL in the five and six months of infection.

### 3.1. Anaplasma Marginale Msp1α Genotype Analysis in the Calves for a Period of a Year

A total of 406 *msp1α* nucleotide sequences were generated from the seven *A. marginale*-positive calves and, in total, 42 unique *msp1α* genotypes were generated from the seven calves over the 12-month study period; however, several of the genotypes occurred in more than one animal (Table 1). Of the total number of *A. marginale* genotypes generated from the seven calves, 76.4% were identified in the five calves at the peri-urban area and 23.6% were identified in the two calves at the wildlife–livestock interface. Calves were infected with four to 13 genotypes (Table 1). Of the 42 *msp1α* genotypes identified, only four occurred in both areas (Table 1). 

The *A. marginale msp1α* genotypes identified in the seven calves were made up of a total of 56 Msp1a repeats; 50 of these have been reported previously while six sequences are novel Msp1a repeats detected for the first time in this study (Figure 4). While only three of the *msp1α* genotypes occurred in both areas, 47.4% of the Msp1a repeats were common to both areas; a further 47.4% of the Msp1a repeats were identified only in calves at the peri-urban area while 5.2% were unique to calves at the wildlife–livestock interface. The six novel Msp1a repeats (named UP37-UP42) were all identified in calves from the peri-urban area. 

### 3.2. Occurrence of A. marginale Multi-Strain Infections in the Calves 

The complexity of *A. marginale* infection in the calves was determined by single/co-infection or superinfection events over the 12-month period (Figure 3). Detection of one or multiple genotypes at the initial time-point was defined as either single or co-infection, respectively. Detection of additional genetically distinct genotypes in the calves over time was defined as superinfection. 

Animals in the peri-urban area acquired four to thirteen *msp1α* genotypes over the 12-month period. At the initial time-point, four of the five calves in the peri-urban area were infected with a single *A. marginale* genotype (or other genotypes were below the level of detection) and one was co-infected with more than two genotypes (Figure 3A). Superinfection with distinct *msp1α* genotypes occurred in all five calves during the study period. The same trend of infection was observed in the two calves that eventually became infected with *A. marginale* at the wildlife–livestock interface (Figure 3B). Although they only became positive for *A. marginale* from time-point T6 and T7, they were either singly infected or co-infected at the beginning but became superinfected with distinct *msp1α* genotypes over time (Figure 3B).

### 3.3. The Composition of Anaplasma spp. in the Ten Calves

PacBio CCS sequencing of 16S rRNA gene amplicons from the final sample taken from each of the ten calves generated a total of 57,683 raw nucleotide sequences that were classified in the genus *Anaplasma* using the RDP 16S classifier. Of these, 55,079 sequences were classified to *Anaplasma* species level using a customized 16S *Anaplasma* NCBI BLASTn database. 

From the 55,079 16S rRNA sequences classified to *Anaplasma* species level, 87% of those were identified in calves at the peri-urban area and only 13% were identified in calves at the wildlife–livestock interface. The raw sequences were randomly sub-sampled to a total of 9950 sequences to equalize the sequencing depth, with 995 sequences analyzed per sample. A total of three *Anaplasma* species were identified in the 10 calves. They consisted mostly of *A. platys*-like 16S rRNA sequences (83.3%), followed by *A. marginale* (16.6%) and *Anaplasma boleense* (<0.1%) as highlighted in Table 2.

*A. marginale* and *A. platys*-like 16S rRNA gene sequences were the most abundant sequences identified in the *Anaplasma* infected calves and frequently occurred as a co-infection. The *A. platys*-like sequence was detected in four of the five calves at the peri-urban area and in three of the calves at the wildlife–livestock interface. Of the three calves that tested negative for *A. marginale* at the wildlife–livestock interface, two were also negative for other *Anaplasma* spp., whilst the third was infected with the *A. platys*-like organism. In terms of *Anaplasma* spp. infections, calf-4 at the peri-urban area that died at T11, was only infected with *A. marginale*. 

### 3.4. 16S rRNA Phylogenetic Analyses

The phylogenetic relationships between *Anaplasma* spp. 16S rRNA gene sequences identified in this study and other published sequences are shown in Figure 5. The phylogenetic tree topologies obtained using three tree algorithms were very similar, and the maximum likelihood tree was chosen as a representative tree. The *A. marginale* sequences had 100% identity to *A. marginale* St Maries [50] and had 99.9% sequence identity to *A. marginale* sequences identified in the various wildlife hosts in the Kruger National Park (KNP) [41]. A minority of sequences had 99.8% identity with *A. boleense* [51], and 99.4% identity with *Anaplasma* sp. KNP9, a novel *Anaplasma* species recently identified in wildlife from KNP [41]. The *A. platys*-like sequences were closely related to *Anaplasma* sp. Omatjenne [52] with 99.7% identity, *A. mymensingh* [53] with 99.9% identity, and “*Candidatus* Anaplasma camelii” [54] with 99.6%. They had 99.7–99.9% identity to *Anaplasma* sp. KNP2, a novel *Anaplasma* species recently identified in wildlife from the Kruger National Park [41].

## 4. Discussion

The presence of *A. marginale* in cattle in the Mnisi community was expected, since the pathogen is currently widespread and endemic in cattle in eight of the nine South African provinces [4,24,25] and is known to occur in most cattle farming areas in the country [10,55]. However, *A. marginale* was detected in only seven of the ten calves in the 12-month study period. Our results further revealed that *A. marginale* infects calves early in their lives or during intra-uterine development [1], since 50% of the calves were infected at T1 and T2, and they did not show clinical symptoms for the duration of the study. This agrees with previous findings [11,12], showing that calves up to 12 months of age are not clinically affected by anaplasmosis. The fact that three of the five calves at the wildlife–livestock interface were not infected was a surprising result. The bovine anaplasmosis cases observed at the wildlife–livestock interface in the Mnisi communal area might thus be attributed to a localised lack of endemic stability since calves at the wildlife–livestock interface are not continually infected with *A. marginale* in their first year when natural immunity is higher. The level of infection (number of infected red blood cells) in the calves that were infected in the two areas did not appear to be significantly different; however, our sample size is very limited and a larger study with more animals would be required to confirm these findings.

Although the Mnisi community is a non-anaplasmosis vaccinating area, absence of *A. centrale* infections was not expected, as *A. centrale* was previously detected in cattle in the study area [33], furthermore the natural circulation *A. centrale* infection was previously observed in buffalo (*Syncerus caffer*), zebra (*Equus quagga burchelli*), warthog (*Phacochoerus africanus*), and lion (*Panthera leo*) in the KNP [56,57,58].

The calves that were infected with *A. marginale* from both areas of the Mnisi community displayed complex *A. marginale* infections driven by co-infection and superinfection, with four to thirteen *msp1α* genotypes detected per animal over the 12-month period, indicating continuous challenge with multiple strains over time that should lead to robust immunity in these animals. Our results are similar to recent findings, where complex *A. marginale* infection with two to six strains per animal was detected in 97% of naïve calves that were introduced into an *A. marginale* infected herd in southern Ghana [31]. Another study from a high *A. marginale* prevalence region in Mexico, showed that up to six *A. marginale* genotypes could be detected per animal using *A. marginale msp1α* genotyping for strain characterization [30]. Although our small sample size might have skewed the results, our findings highlight differences in temporal *A. marginale* infection dynamics between the villages, with all five of the calves at Eglington village (a peri-urban area) being infected at T1 and T2, but only two of the three calves at Utha A (at the wildlife livestock–interface) infected at T5 and T6, and no infection detected in the remaining three calves (one at Utha A and two at Dixie at the wildlife–livestock interface). Factors such as cattle density and management, which differ at the three villages, may drive the dynamics of *A. marginale* infection, with a lack of early *A. marginale* infection at the wildlife–livestock interface resulting in the frequent clinical cases in the area. The rapid migration of *R. microplus* ticks (larvae and adult ticks) from infested to un-infested cattle has been implicated in the interstadial transmission of *A. marginale* [59]; furthermore, attachment of three infected *R. microplus* ticks is sufficient for transmission of *A. marginale* from infected to naïve cattle [60], while a single *Dermacentor andersoni* infected tick is sufficient for transmission [61]. Therefore, transmission of *A. marginale* is more likely to occur in areas where cattle density is higher, due to increased opportunities for migration of vector ticks from *A. marginale*-infected to uninfected cattle, thus increasing the chances of transmission in the area. Additionally, different methods of acaricide treatments are used in the two study areas, and this might have had an effect on the disease transmission dynamics observed. The frequency of cattle dipping in the Mnisi communal area is greatly affected by water shortages. Cattle in the peri-urban site, Eglington village, as well as Utha A, at the wildlife–livestock interface are dipped using the plunge method of dipping cattle, while a hand spraying method is used at Dixie village (at the wildlife–livestock interface). Several factors, such as the inability to clean and empty the dip tank resulting in a heavily silted dip tank, and incorrect mixing ratios of water and the acaricide, have been shown to be the prime causes of tick control failure at communal plunge dip tanks [62,63] such as the ones used at Eglington and Utha A. Thus, the hand spraying method, as is used at Dixie village, may be more effective in controlling tick infestation and thus preventing disease transmission, than the plunge method of cattle dipping where the concentration of the acaricide in the dip tank might not be consistent. 

Our findings further indicate the presence of other *Anaplasma* species circulating in the calves, which mainly comprised an *A. platys*-like organism that is closely related to a novel *Anaplasma* species recently identified in wildlife in the Kruger National Park [41]. Very low levels of *A. boleense* 16S rRNA sequences were also detected in the calves, which were also previously detected in cattle in the area [33]. The high levels the *A. platys*-like organism present in the calves suggest the presence of ticks responsible for the transmission of this organism in the area. It has been postulated that exposure to closely related non-pathogenic organisms might provide some cross-protection against the pathogenic species in cattle and thus decrease the pathogenicity of the infection [64]. It is thus possible that infection with the *A. platys*-like organism might confer heterologous protection against *A. marginale* in cattle in the area, thus contributing to endemic stability of *A. marginale*. An experimental study conducted in Kenya [65] showed that cattle are highly susceptible to infection by less pathogenic *Anaplasma* species from wildlife hosts. Cattle having recovered from anaplasmosis caused by *Anaplasma* species from wildlife showed slight protection against subsequent infection with *A. marginale*. Future studies should be aimed at confirming these observations and further determining the mechanisms underlying heterologous protection against bovine anaplasmosis by closely related non-pathogenic species. 

## 5. Conclusions

Complex *A. marginale* infection in the Mnisi community is driven by co-infection and superinfection. Factors such as cattle density and management, which differ at the three villages, may drive the temporal dynamics of the infection. A localized lack of endemic stability at the wildlife–livestock interface could result in clinical cases caused by challenge with *A. marginale* at a later point in life. Our findings suggest that cattle in the Mnisi community are exposed to other *Anaplasma* spp. which might confer cross-protection against the pathogenic *A. marginale* infection and might suggest that other, previously unrecognized *Anaplasma* species could contribute to the control of bovine anaplasmosis in South Africa. While our results suggest that there are differences in the time-course of infection in calves in different areas of the Mnisi community, it should be noted that only five calves were examined from each area. A future in-depth longitudinal study in more villages of the Mnisi community with a larger sample size is recommended to confirm and further analyze the dynamics of *A. marginale* infections in the Mnisi communal area, especially at the wildlife–livestock interface.

## Figures and Tables

**Figure 1 microorganisms-11-00465-f001:**
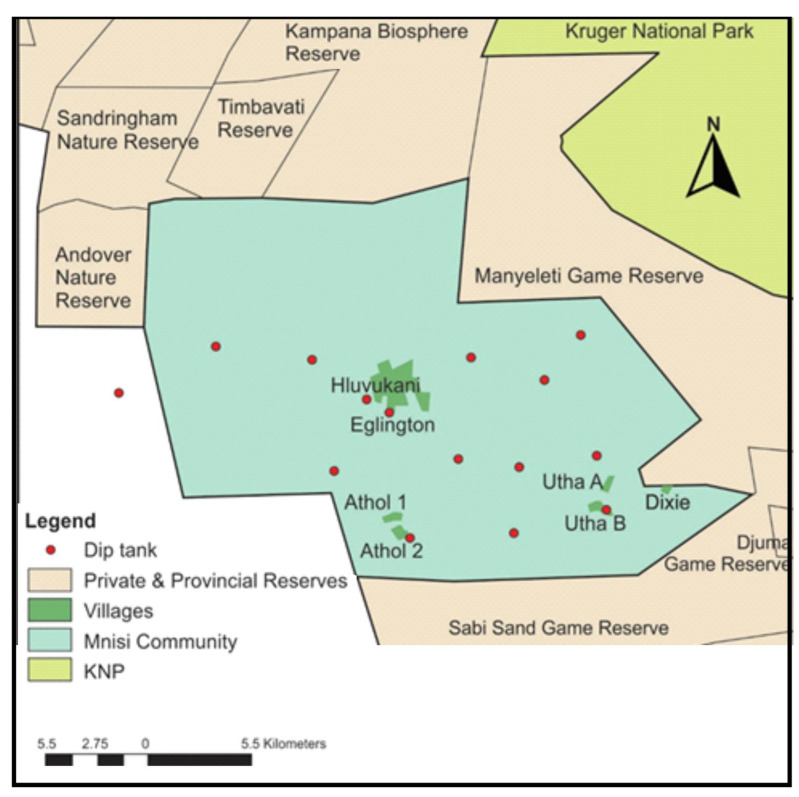
Map of the Mnisi communal area, showing the location of the three villages where the study was conducted, Eglington, Utha A, and Dixie, relative to various wildlife reserves and the Kruger National Park (KNP).

**Figure 2 microorganisms-11-00465-f002:**
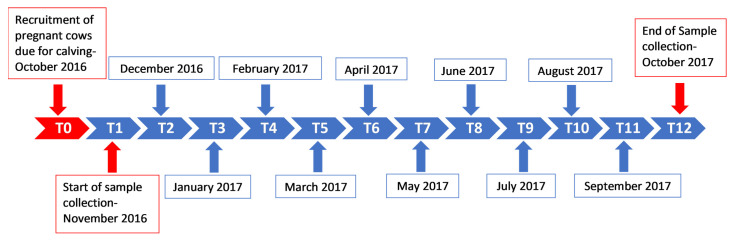
Sample collection timeline for the study. Samples were collected monthly from the ten calves for a period of one year (T1—November 2016, T2—December 2016 and T12—October 2017), from the ages of 0–1 months old (0–1 M) to 11–12 months old (11–12 M). T(x) = time point (month number).

**Figure 3 microorganisms-11-00465-f003:**
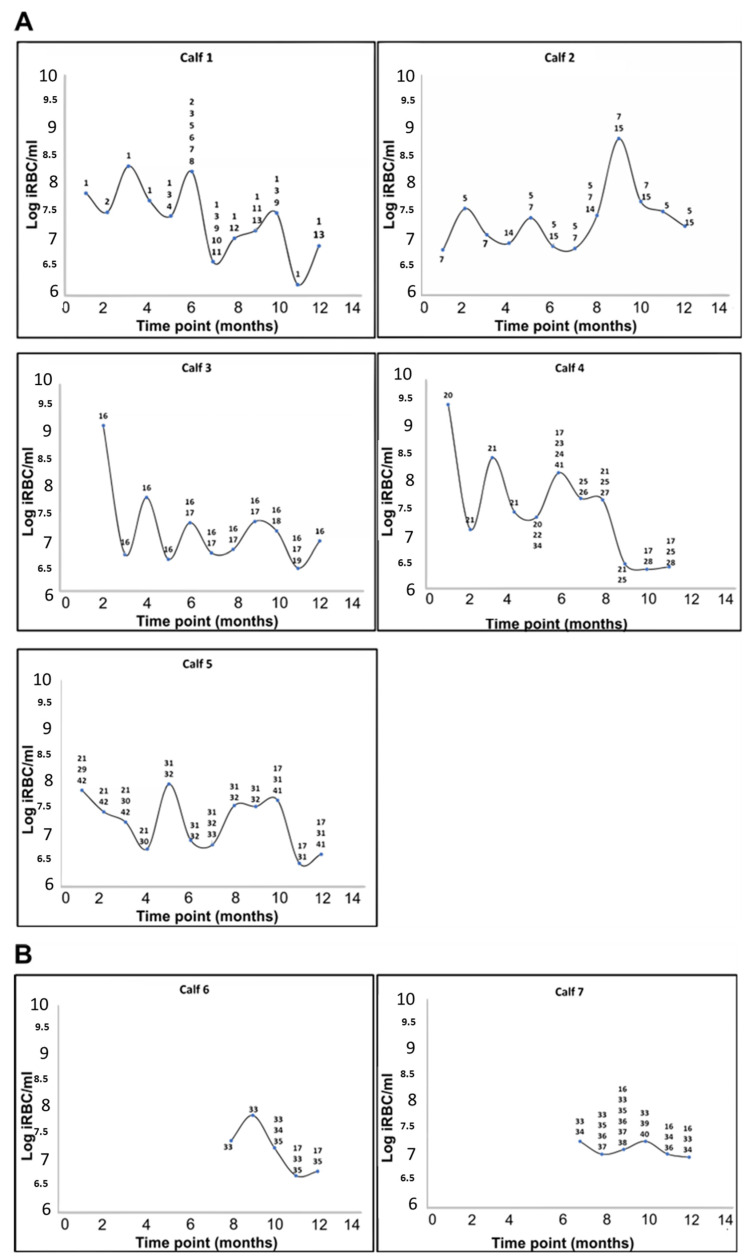
Cyclic *A. marginale* rickettsaemia in calves from the Mnisi community as determined by qPCR [25,34]. The level of infection is expressed as the log of infected red blood cells (RBC) per mL (iRBC/mL) of blood. (**A**) Calves infected with *A. marginale* at the peri-urban area. (**B**) Calves infected with *A. marginale* at the wildlife–livestock interface. The number and type of *A. marginale msp1α* genotypes detected at each time-point are indicated for each calf; genotypes were assigned numbers as shown in Table 1.

**Figure 4 microorganisms-11-00465-f004:**
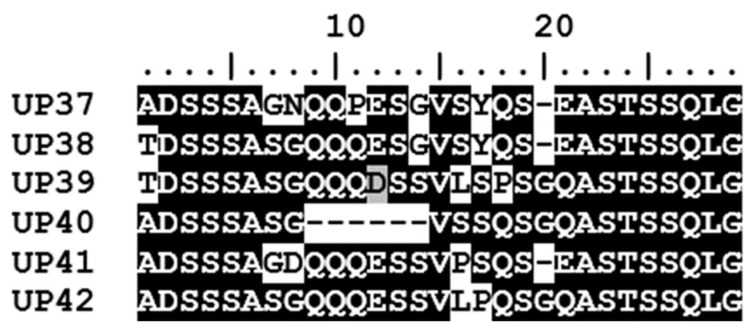
Novel Msp1a repeat sequences detected in this study. Six novel Msp1a repeats (UP37-UP42) were identified in the five calves located in the peri-urban area of the Mnisi community, Mpumalanga. Msp1a sequences were aligned using BioEdit. Conserved amino acid residues in the alignment are highlighted by white text on a black background, while variable residues are shown by black text on a white background.

**Figure 5 microorganisms-11-00465-f005:**
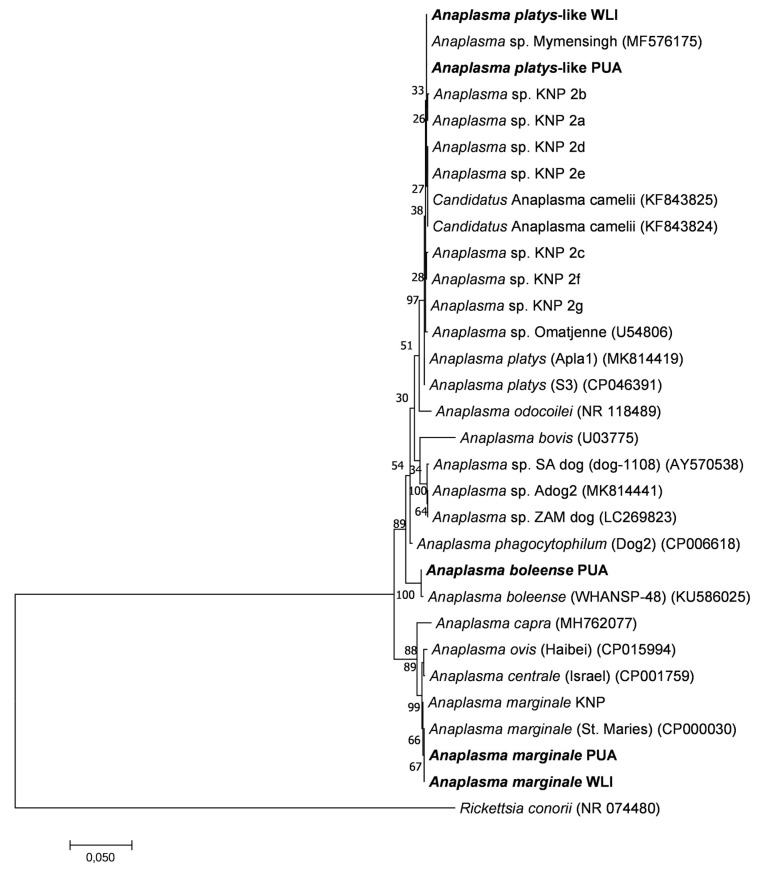
Maximum likelihood phylogenetic tree showing the phylogenetic relationships between *Anaplasma* 16S rRNA gene sequences obtained from the ten calves included in this study (in bold) and previously published *Anaplasma* 16S rRNA gene sequences. Near full-length 16S rRNA gene sequences of approximately 1328 bp in length were used to construct the tree. The numbers associated with each node indicate the percentage of 1000 bootstrap replications supporting the node. Phylogenetic analyses were conducted in MEGA7 with an HKY85 evolutionary model. Sequences with abbreviations PUA, WLI, and KNP highlight that the sequences were retrieved from animals in the peri-urban area, wildlife–livestock interface and Kruger National Park, respectively. *Rickettsia conorii* was used as the outgroup. The scale bar highlights a 5% nucleotide sequence divergence.

**Table 1 microorganisms-11-00465-t001:** *Anaplasma marginale msp1α* genotypes identified from infected calves over the 12-month study period.

Calf No.^a^	No. of Genotypes	Size (bp)	No. of Msp1a Repeats	Genotype	Number Allocated to Genotype	Genotype #, If Previously Detected in Study
1 (EG1)	13	949	5	171-2;UP3 ^b^ 172-2;UP4 61 172-2;UP4 172-2;UP4	1	
		700	2	τ 10	2	
		697	2	171-2;UP3 172-2;UP4	3	
		866	3	171-2;UP3 172-2;UP4 61	4	
		836	6	UP37 ^c^ UP31 UP31 UP31 UP31 UP31	5	
		781	2	τ UP31	6	
		787	3	τ 10 22-2	7	
		610	1	UP38 ^c^	8	
		893	5	61 172-2;UP4 61 172-2;UP4 172-2;UP4	9	
		781	3	61 172-2;UP4 169-2	10	
		781	3	61 172-2;UP4 172-2;UP4	11	
		781	3	171-2;UP3 172-2;UP4 172-2;UP4	12	
		697	2	172-2;UP4 172-2;UP4	13	
2 (EG2)	4	784	3	UP39 ^c^ 10 UP31	14	5, 7
		781	3	179-2 169-2 172-2;UP4	15	
3 (EG3)	4	787	3	84 172-2;UP4 172-2;UP4	16 ^d^	
		959	5	34 3 36 36 38	17 ^d^	
		958	5	13 27 36 3 38	18	
		700	2	13 27	19	
4 (EG4)	12	1040	6	34 36 36 3 36 38	20	17 ^d^
		1037	6	UP40 ^c^ β β β β F	21	
		959	5	34 36 36 27 18	22	
		954	5	MZ2 3 UP41 ^c^ 36 38	23	
		880	4	3 β 36 3	24	
		962	5	42 43 43 25 31	25	
		1131	7	34 3 UP1 43 43 25 31	26	
		1026	6	UP40 ^c^ β β β Is9;78 31	27	
		965	5	84 172-2;UP4 172-2;UP4 172-2;UP4 172-2;UP4	28	
		870	4	τ 22-2 13 18	34 ^b^	
		705	2	34 3	41	
5 (EG5)	9	689	2	UP40 ^c^ β	29	17 ^d^, 21, 41
		1001	5	UP40 ^c^ β β β F	30	
		875	4	42 43 25 31	31	
		790	3	42 UP42 ^c^ 27	32	
		791	3	H M 27	33 ^c^	
		602	1	UP40 ^c^	42	
6 (UT1)	4	919	5	UP5 UP6 25 31 31	35	17 ^d^, 33 ^d^, 34 ^d^
7 (UT2)	9	1075	7	UP5 UP6 25 31 UP6 27 18	36	16 ^d^, 33 ^d^, 34 ^d^, 35
		787	3	84 61 31	37	
		863	4	UP5 UP6 25 31	38	
		1202	8	UP5 UP6 25 31 UP6 25 31 31	39	
		955	4	84 Is9;78 31 31	40	

^a^ Calves 1–5 were in the peri-urban area, Eglington; Calves 6–7 were in the wildlife–livestock interface, Utha A and Dixie. ^b^ Msp1a repeats denoted with a semicolon (e.g., 171-2;UP3) have been given two names in the literature. ^c^ indicates a novel Msp1a repeat (red). ^d^
*msp1α* genotypes that occur in both areas.

**Table 2 microorganisms-11-00465-t002:** The percentage of 16S rRNA sequences of each *Anaplasma* spp. identified.

Calf No.	*A. platys*-like	*A. marginale*	*A. boleense*
1 (EG1)	67.3	32.3	0.4
2 (EG2)	98.7	1.3	0
3 (EG3)	56.0	43.9	0.1
4 (EG4)	0	100	0
5 (EG5)	68.5	31.3	0.2
6 (UT1)	7.0	93.0	0
7 (UT2)	95.3	4.7	0
8 (DI1)	0	0	0
9 (DI2)	0	0	0
10 (DI3)	100	0	0

## Data Availability

All the sequences generated in this study are available under BioProject accession number PRJNA929355. *Anaplasma msp1α* and 16S rRNA nucleotide sequences generated in this study were deposited in GenBank under accession numbers OQ384772–OQ384912 and OQ348128-OQ348132 respectively. The raw microbiome data from the ten calves is available at the Sequence Read Archive (SRA) with BioProject accession number PRJNA929355.

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
