# Peer review of "Temporal Dynamics of Anaplasma marginale Infections and the Composition of Anaplasma spp. in Calves in the Mnisi Communal Area, Mpumalanga, South Africa"

_microorganisms, 2023, doi:10.3390/microorganisms11020465_

Round 1

Reviewer 1 Report

S. Marcus Makgabo and colleagues submitted a paper titled "Temporal dynamics of Anaplasma marginale infections and the composition of Anaplasma spp. in calves in the Monks communal area, Mpumalanga, South Africa" for consideration in Microorganism. The aim of the study was to investigate the species and phylogenetic diversity of Anaplasma in rural areas in South Africa in blood samples collected from calves.

In the introduction, the authors summarize the issue of Anaplasma infections in cattle and explain why their research is important. I agree with this argument, however, I have a few comments on the introduction, which I have included in the pdf file attached to this review.

The methods are almost correctly described, however, I believe that the criteria for selecting sequences in BLAST used to construct the phylogenetic tree should be included in the section of the methodology. Why did the authors decide to choose the ML method for trees construction?

The authors report that the study included animals from the time of birth for a year period. In my opinion, the results should include a table showing the prevalence of Anaplasma in individual animal. 

In the text of ms, the authors use the terms microbiome and pathogens interchangeably. I suggest using pathogens only, although I guess it's because of on A. paltys?

Please see my detailed comments in pdf (text highlighted in yellow is often due to the lack of italics) 

Reviewer 2 Report

In "temporal dynamics of anaplasma marginale infections and the composition of anaplasma spp. in calves in the Mnisi communal area, Mpumalanga, South Africa" Makgabo et al describe a longitudinal observational study, where anaplasma spp presence was investigated using qPCR and 16S rRNA sequencing. The article is very well written, and the study is very interesting, clearly demonstrating a need for a larger study to confirm the findings. 

The authors describe the methods and results in a clear and concise manner, with adequate illustrations. I have a couple of questions that would enhance the impact of the article, if they are addressed. 

1) With the selection of the cattle, three different areas were selected. Can the authors elaborate further on how much cross-traffic there is between the different areas, either by the cattle, or humans (trading, veterinarians, travel etc). This would be interesting to those not familiar with the area, and can also allude to possible influences on spread of disease. 

2) In the conclusion, the authors mention that the lack of endemic anaplasma in the cattle in the wild-life interface could lead to more severe disease, especially since calves would not be exposed in their first year of life. This statement made me curious to the disease course in the different areas, both during the study and historically, and how does it compare to anaplasma copy numbers that can be detected in the blood? Can the authors give some more information on that in the results / discussion section?

3) Can the authors describe the accuracy of the sequencing in more detail? What was the read depth for each sequence, and was this consistent throughout all samples, and how clean were the sequences? with the detection of new sequences, this information is useful to include. 

4) With the isolation of new sequences, do the authors have any information on the strength of the sequences, are the mutations beneficial to anaplasma, do they influence stability / infectivity / survival of the pathogen, do they affect the disease course? 

5) Regarding the different subtypes detected in the calves over time, and sometimes sequences not being detected, do the authors have any idea why this is? Is this purely read-depth, and too low to detect, or (like my previous question) are some subtypes stronger than others (ie have you done an analysis comparing copy numbers of each subtype comparing to others at the same time point, and then on a temporal level?). 

6) It seems that there is a very higher variability in the number of subtypes found, and especially with such a small sample size, can the authors comment whether this wide variety was expected, and whether it is expected to expand further with a larger sample size, or that you expect to find more of the same?

I read the manuscript with great interest, and look forward to reading the answers to these questions.

Round 2

Reviewer 1 Report

The authors provided detailed replies to my comments. The manuscript has been revised, and in my opinion can be considered for publication